# High-Performance Resistive Switching in Solution-Derived IGZO:N Memristors by Microwave-Assisted Nitridation

**DOI:** 10.3390/nano11051081

**Published:** 2021-04-22

**Authors:** Shin-Yi Min, Won-Ju Cho

**Affiliations:** Department of Electronic Materials Engineering, Kwangwoon University, 20, Gwangun-ro, Nowon-gu, Seoul 01897, Korea; kkuregi1234@naver.com

**Keywords:** memristor, IGZO:N, microwave annealing (MWA), microwave-assisted nitridation, synaptic weight modulation

## Abstract

In this study, we implemented a high-performance two-terminal memristor device with a metal/insulator/metal (MIM) structure using a solution-derived In-Ga-Zn-Oxide (IGZO)-based nanocomposite as a resistive switching (RS) layer. In order to secure stable memristive switching characteristics, IGZO:N nanocomposites were synthesized through the microwave-assisted nitridation of solution-derived IGZO thin films, and the resulting improvement in synaptic characteristics was systematically evaluated. The microwave-assisted nitridation of solution-derived IGZO films was clearly demonstrated by chemical etching, optical absorption coefficient analysis, and X-ray photoelectron spectroscopy. Two types of memristor devices were prepared using an IGZO or an IGZO:N nanocomposite film as an RS layer. As a result, the IGZO:N memristors showed excellent endurance and resistance distribution in the 10^3^ repeated cycling tests, while the IGZO memristors showed poor characteristics. Furthermore, in terms of electrical synaptic operation, the IGZO:N memristors possessed a highly stable nonvolatile multi-level resistance controllability and yielded better electric pulse-induced conductance modulation in 5 × 10^2^ stimulation pulses. These findings demonstrate that the microwave annealing process is an effective synthesis strategy for the incorporation of chemical species into the nanocomposite framework, and that the microwave-assisted nitridation improves the memristive switching characteristics in the oxide-based RS layer.

## 1. Introduction

Over the past few decades, information science technology and electronic applications have required versatile electrical units, computing methods, and materials [1,2]. To overcome the von Neumann bottleneck issue, which mainly occurs due to the physically separated processing and memory units, innovative electronic devices that can replace conventional CMOS technology and electromagnetic passive circuit elements such as resistors, inductors, and capacitors have been researched [3,4]. In particular, “Beyond CMOS”-like high computing efficiency, ultra-high density, and low-power consumption properties are essential for the emerging big-data era [5]. Accordingly, memristors, which are nonlinear two-terminal electrical components, and were theoretically proposed in 1971, have attracted significant attention in recent years because of their exceptional features and advantages [6]. The metal/insulator/metal (MIM) sandwich structure is geometrically simple, and these memristor devices possess promising features such as high scalability, low-power consumption, nonvolatile data storage, and multi-resistance state controllability. The resistance state of memristors can be gradually modulated in the insulator layer, and various resistive switching (RS) methods such as ferroelectric switching, phase-change switching, and oxide-based switching have been reported [7,8,9]. Among these methods, the oxide-based RS behaviors can be realized in a simple material combination, and they exhibit characteristics such as stable mechanical strength, thermal stability, and symmetrical resistance modulation, contrary to ferroelectric-based or phase-change-based switching [10,11]. Therefore, a wide variety of oxide-based materials have been researched to improve the memristive switching characteristics. In particular, the multi-component metal-oxide (MeO_x_)-based materials help form stable conductive filaments (CFs) by controlling the chemical composition ratio, process ambient, and annealing method [12,13]. 

In this study, we proposed two-terminal memristor devices using a solution-derived In-Ga-Zn-oxide (IGZO) nanocomposite as a MeO_x_-based RS layer, and we evaluated the memristive switching characteristics. However, the IGZO-based RS devices grown by the sputtering system have been studied and improved through the conventional thermal annealing (CTA) process under a nitrogen-ambient [14]. Compared to the previous study, solution-based films are promising for versatile electronic applications, and the microwave annealing (MWA) technique can offer a more effective annealing treatment than the CTA process through its rapid volumetric heating by the conversion of electromagnetic energy into thermal energy [15,16]. The primary aim of this study was the use of the MWA process to eliminate the moisture and stabilize the solution-derived IGZO nanocomposite films. Furthermore, the MWA-assisted synthesis strategy can be potentially used for the incorporation of chemical species in the nanocomposite framework in a short process time [17,18]. Therefore, the microwave-assisted nitrogen synthesis effect on solution-derived IGZO nanocomposites was systematically evaluated through the chemical solution etching test, optical absorption coefficient analysis, and X-ray photoelectron spectroscopy (XPS). As a result, the improved memristive switching properties and synaptic weight modulation characteristics of the memristor devices were analyzed using nitride IGZO (IGZO:N) nanocomposites as an RS layer compared with IGZO films. 

## 2. Experimental Methods

### 2.1. Material Specifications

The material specifications of the study are as follows: p-type Si substrate (plane, (100); resistivity range, 1–10 Ω∙cm; LG Siltron Inc., Gumi, Korea), Ti pellet (purity > 99.999%; THIFINE Corp., Incheon, Korea), Pt pellet (purity > 99.95%; THIFINE Corp., Incheon, Korea), indium (III) nitrate hydrate (In(NO_3_)_3_∙xH_2_O; purity > 99.99%; Sigma Aldrich, Saint Louis, MO, USA), gallium (III) nitrate hydrate (Ga(NO_3_)_3_∙xH_2_O; purity > 99.99%; Sigma Aldrich, Saint Louis, MO, USA), zinc acetate dehydrate (Zn(CH_3_COO)_2_∙2H_2_O; purity > 99.99%; Sigma Aldrich, Saint Louis, MO, USA), 2-methoxyethanol (purity > 99.99%; Sigma Aldrich, Saint Louis, MO, USA), and ethanolamine (purity > 99.5%; Sigma Aldrich, Saint Louis, MO, USA). 

### 2.2. IGZO Solution Synthesis Procedure 

The precursor for the solution-derived IGZO nanocomposite was synthesized through a sol–gel reaction. The IGZO precursor solution was prepared as follows: a powder of indium (III) nitrate hydrate, gallium (III) nitrate hydrate, and zinc acetate dehydrate in a 1:3:1 molar ratio was dissolved in a mixture of 2-methoxyethanol (20 mL) and ethanolamine (2.5 mL) solutions. Further, the powders and solution mixture were contained in a closed vessel, and they were synthesized by using a constant magnetic stirring system at 800 rpm for 2 h at 60 °C. Then, the 1:3:1 molar-ratio IGZO solution was aged for 24 h at room temperature (25 °C) and finally filtered through a 0.2 μm-pore-size polytetrafluoroethylene (PTFE) membrane syringe filter (Whatman International Ltd., Maidstone, UK) for further purification.

### 2.3. IGZO:N Memristor Devices Fabrication

To fabricate solution-derived IGZO:N memristors with microwave-assisted nitridation, p-type Si substrates with a thermally grown 300 nm-thick oxide were used as the starting materials. The p-type Si substrates were cleaned with a wet-chemistry-based standard Radio Corporation of America method. For the bottom electrode (BE) of the MIM structure memristor devices, 10 nm-thick Ti and 100 nm-thick Pt thin films were sequentially deposited on the substrate through an electron-beam (E-beam) evaporator system. Further, the solution-derived IGZO:N nanocomposites RS layer, which is the most significant part for memristor operations, was formed as follows: The synthesized 1:3:1 molar-ratio IGZO solution was spin-coated on the BE layer at 500 rpm for 10 s, followed by 2000 rpm for 30 s. Then, for the microwave-assisted IGZO:N nanocomposites synthesis, the MWA process was conducted at a frequency of 2.45 GHz and a power of 1800 W for 2 min under a nitrogen-ambient. Finally, a circular-shaped Ti top-electrode (TE) with a diameter size of 200 μm was deposited on the RS layer at a thickness of 100 nm using an E-beam evaporator system and a shadow mask. In order to evaluate the microwave-assisted nitridation effect on the memristor operations, the MWA-treated memristor devices under an air-ambient (IGZO memristor devices) were also fabricated and systematically compared with IGZO:N memristor devices. 

### 2.4. Characterization of IGZO:N Memristor Devices

The fabricated IGZO:N and IGZO memristor devices were positioned at a two-point probe system within a dark chamber to evaluate the memristive synaptic behaviors without light and electrical noise, respectively. The electrical characteristics were analyzed with an Agilent 4156B Precision Semiconductor Parameter Analyzer (Hewlett-Packard Corp., Palo Alto, CA, USA). The electrical synapse pulses were applied through an Agilent 8110A Pulse Generator (Hewlett-Packard Corp., Palo Alto, CA, USA). The optical microscope image analysis for the fabricated memristor devices was proceeded with a magnification of 150× by using an SV-55 Optical Microscope (SOMETECH, Seoul, Korea). The optical transmittances of RS layers were evaluated in the wavelength spectra range of 190–1100 nm by using an Agilent 8453 Ultraviolet-Visible Spectrophotometer (Hewlett-Packard Corp., Palo Alto, CA, USA). The thicknesses of RS layers were measured by a DetakXT Bruker stylus profiler (Bruker Corp., Billerica, MA, USA). In addition, the chemical composition of the IGZO:N and IGZO nanocomposites were analyzed by XPS (PHI 5000 Versa Probe II, ULVAC, Chigasaki, Kanagawa, Japan), which was conducted using monochromatic Al-Kα radiation (λ = 0.833 nm).

Figure 1a depicts a schematic diagram of the proposed solution-derived IGZO:N memristors by microwave-assisted nitridation, and Figure 1b,c show the optical microscope images (150× magnification) of the memristor devices with MWA treated under an air-ambient (IGZO) and nitrogen-ambient (IGZO:N), respectively. Figure 1d represents a simple structure of the biological synapse, and Figure 1e provides a simplified mechanism of conductance modulation through the CFs formation/rupture process in IGZO:N memristor devices. 

## 3. Results and Discussion

### 3.1. Microwave-Assisted Nitridation Effect

The IGZO thin film is a well-known n-type oxide-semiconductor that consists of ionically bonded heavy metal cations with an electronic configuration of (*n* − 1)*d*^10^*ns*^0^ (*n* ≥ 4) [19,20]. The electron transport paths in the conduction band (*E_c_*) of IGZO nanocomposites mainly consist of overlapping spherical In 5 *s*-orbitals, and the free carriers can be provided from defects such as oxygen vacancies and metal ion interstitials [21,22]. 

To investigate the effects of microwave-assisted nitridation on solution-derived IGZO nanocomposites, the IGZO and IGZO:N films that were formed on the p-type Si substrate were dipped in a buffered-oxide-etch (BOE; 30:1) solution for 30 s. In Figure 2a, the IGZO film was etched by the BOE treatment and the thickness of the initial IGZO film decreased. On the other hand, the IGZO:N film was not etched by BOE treatment, which indicates the difference in the chemical compositions of IGZO and IGZO:N nanocomposites. Figure 2b represents the optical transmittance spectra of the IGZO and IGZO:N films that were formed on the glass substrate (7059 glass; Corning Inc., New York, NY, USA). The inset depicts the optical transmittance at the visible-light wavelength (400–700 nm) region and the average transmittance values. The average optical transmittances of IGZO and IGZO:N films at the visible-light wavelength region were 90.2 and 80.4%, respectively. The optical absorption coefficients (α) of IGZO and IGZO:N films that were calculated from the transmittance spectra are represented in Figure 2c. The absorption coefficients were extracted from the ultraviolet wavelength region of optical transmittance spectra and were calculated by the following equation neglecting the reflection coefficient [23]: $$\alpha =\frac{1}{t}ln\left(\frac{1}{T}\right)$$
where ***t*** is the film thickness and ***T*** is the ratio of transmittance. The optical energy bandgap (***E_g_***) can be extracted through the following relationship of the absorption coefficient:$$\alpha \;\propto \;{\left(h\nu -E_{g}\right)}^{1/2}$$
where ***hν*** is the photon energy. The ***E_g_*** values of IGZO and IGZO:N nanocomposites were 3.82 and 3.68 eV, respectively. The average optical transmittance and ***E_g_*** of IGZO:N nanocomposites decreased by the microwave-assisted nitridation process. In the fundamental bandgap features of IGZO nanocomposites, the conduction band (*E_c_*) was mainly composed of the metal cation-related *s*-orbital, especially In 5 *s*-orbitals, and the highest valence band (*E_v_*) mainly formed the oxygen-related *2p*-orbital. The oxygen vacancies resulted in a deep sub-gap state above the *E_v_* and were sensitive to the annealing process [24]. When nitrogen was incorporated into the IGZO by an annealing process under a nitrogen ambient, the nitrogen atoms partially occupied the oxygen vacancies and bonded with the inactive interstitial oxygen. As the nitrogen-related *2p*-orbitals have a higher potential energy than the oxygen-related *2p*-orbital, the bandgap of IGZO:N nanocomposites became narrower than that of IGZO [25,26]. As a result, the IGZO:N had a lower optical transmittance than IGZO through a smaller energy bandgap, which demonstrates that the microwave-assisted nitridation process was successfully implemented on the IGZO:N nanocomposites.

### 3.2. Chemical Compositions of IGZO:N Nanocomposites

Figure 3 shows the chemical compositions of IGZO and IGZO:N nanocomposites through the XPS spectra evaluations. The XPS spectra were measured after the surface was etched with Ar^+^ ion-etching for 1 min, in order to analyze the chemical compositions by avoiding any surface contamination. To investigate the microwave-assisted nitridation effects on solution-derived IGZO nanocomposites, each metal species was deconvoluted into three individual component peaks: metal–metal bonds, metal–nitrogen bonds, and metal–oxygen bonds. In addition, the O *1s* peak was also deconvoluted into the stoichiometric oxygen (M-O) bonds, oxygen-vacancies (M-O_vac_) bonds, and oxygen impurities (M-OH) bonds such as H_2_O, CO_3_, or chemisorbed oxygen [27]. 

In Figure 3, the deconvoluted core-level spectra of In *3d*_3/2_, Ga *2p*_3/2_, and Zn *2p*_3/2_ show that the atomic concentration of metal-N bonds increased and the metal-O bonds decreased in the IGZO:N nanocomposites. Furthermore, in the O *1s* spectra, the atomic concentration of M-O_vac_ bonds increased from 14.6% (in IGZO nanocomposites) to 35.0% (in IGZO:N nanocomposites) because of the microwave-assisted nitridation effects. This significant evidence indicates the incorporation of nitrogen species into the IGZO nanocomposites by the MWA process, which could result in sufficient oxygen-vacancies in the RS layer [12,28,29]. Table 1 and Table 2 list the total binding energy values and atomic concentrations of individual component peaks in IGZO and IGZO:N nanocomposites. 

### 3.3. Resistive Switching Operations and Stability Evaluation

Figure 4a represents the current–voltage (*I–V*) characteristic curves of the fabricated IGZO and IGZO:N memristor devices. The sequential DC voltage loop of 0 V⟶1.5 V⟶0 V⟶−1.0 V⟶0 V (in 0.05 V step) was applied to the Ti-TE, and the current flow through the RS layer was measured when the Pt-BE was grounded, as shown in Figure 1a. 

In the positive-voltage region (1), the current through the RS layer increased with the voltage. The negatively charged oxygen ions in the RS layer could drift to the Ti-TE/RS interfacial layer because of the positive electric field. Thus, the localized CFs by oxygen-vacancies were formed in the RS layer, which further triggered the conversion from the high-resistance state (HRS) to the low-resistance state (LRS). This electrical phenomenon is called the “set process” and is depicted in Figure 1e. Conversely, in the negative-voltage region (4), the oxygen ions from the Ti-TE/RS interfacial layer or around the CFs diffused back to the localized CFs because of the negative electric field and neutralized the oxygen-vacancies. Therefore, the CFs were ruptured at the negative-voltage region, and the resistance state was modulated from the LRS to the HRS (“reset process”) [30,31,32]. In Figure 4a, both the IGZO and IGZO:N memristor devices exhibited bipolar RS (BRS) operations wherein the set/reset processes occurred in opposite polarity, and the IGZO:N memristor showed a larger resistance difference in the LRS and HRS. Figure 4b shows the cumulative resistance distribution under 10^2^ sequential DC cycles of the IGZO and IGZO:N memristor devices. The resistance values of LRS and HRS were extracted from a read voltage of 0.1 V in each BRS cycle. As a result, the IGZO:N memristor devices had a more stable resistance distribution and a larger resistance difference in the LRS and HRS compared to the IGZO memristor devices. 

In Figure 5, the repeated DC BRS endurance test was conducted for IGZO memristors and IGZO:N memristors for further stability investigation. 

The BRS operations occurred repetitively following the DC voltage sweep direction from (1) to (4). Figure 5a represents the BRS *I–V* characteristic curves from the 1st cycle to the 10^2^th cycle for IGZO memristors, which irregularly switched during repeated cycles. Conversely, Figure 5d shows the BRS *I–V* characteristic curves from the 1st cycle to the 10^3^th cycle for IGZO:N memristors, which exhibited highly stable switching operations as the cycles repeated. In the repeated BRS *I–V* curves, the resistance values of the HRS and LRS were extracted at a read voltage of 0.1 V, and they have been depicted in Figure 5b for IGZO memristors and in Figure 5e for IGZO:N memristors. In IGZO memristors, each resistance value changed irregularly and the average values with standard deviations (SDs) of the HRS and LRS were 4.02 × 10^2^ Ω (SD: 7.52 × 10^2^ Ω) and 6.70 × 10^1^ Ω (SD: 1.04 × 10^1^ Ω), respectively. In IGZO:N memristors, each resistance value was stably repeated and the average values with standard deviations (SD) of the HRS and LRS were 8.30 × 10^2^ Ω (SD: 3.40 × 10^1^ Ω) and 8.95 × 10^1^ Ω (SD: 2.01 × 10^0^ Ω), respectively. Figure 5c,f show the cumulative distribution of the set voltage (*V_set_*) and reset voltage (*V_reset_*) for BRS operation of the IGZO and IGZO:N memristor devices, respectively. The insets indicate the calculated BRS operation power for the set process (*P_set_*) and reset process (*P_reset_*). The *P_set_* and *P_reset_* can be defined as *P_set_* = *V_set_* × *I_cc_* and *P_reset_* = |*V_reset_* × *I_reset_*|, respectively. Here, the compliance current (*I_cc_*) is the current limitation that prevents a hard breakdown during the set process, and the reset current (*I_reset_*) is the peak current during the reset process [33]. Therefore, the average values of *V_set_*, *V_reset_*, *P_set_*, and *P_reset_* were 0.71 V, −0.74 V, 8.23 mW, and 7.06 mW for the IGZO memristors, and 0.78 V, −0.67 V, 7.82 mW, and 5.95 mW for IGZO:N memristors, respectively. The total BRS operation parameters evaluated by the endurance test are listed in Table 3 for IGZO and IGZO:N memristor devices. Thus, the IGZO:N memristor devices represent a highly improved operation stability with small standard deviations during repeated DC BRS endurance evaluation, as compared to IGZO memristors. These beneficial switching behavior stabilization effects are due to the oxygen-vacancies-rich CFs in IGZO:N nanocomposites, which are attributed to microwave-assisted nitridation [12,34,35]. 

### 3.4. Current Conduction Mechanism of IGZO:N Memristor

To investigate the current conduction mechanism for IGZO:N memristor devices, the set and reset processes in BRS *I–V* characteristic curves were analyzed with double-logarithmic plotting and are represented in Figure 6. In the positive and negative bias sections of Figure 6, the HRS in *I–V* curves can be divided into three distinct regions by a noticeable slope difference: ohmic conduction (*I*∝*V*, blue line) in region (1), trap-filled-limited conduction (*I*∝*V^2^*, red line) in region (2), and the trap-controlled space-charge-limited conduction (SCLC) (a steep increase in *I*, green line) in region (3). 

When a low positive bias was applied in region (1) of Figure 6b, the injected carrier density was insufficient, compared to the number of thermally generated free charge carriers in the RS layer, and the *I–V* curve followed the ohmic conduction-related mechanism (slope ~ 1) [36]. As the applied bias increased above the transition voltage (*V_tr_*) in region (2), the density of injected carriers increased, compared to the thermally generated free charge carriers. As the localized CFs in IGZO:N nanocomposites were mainly composed of oxygen-vacancies like defect-related traps, the current nonlinearly increased following the trap-filled-limited conduction (slope ~ 2) [37,38]. When the applied bias overcame the trap-filled-limited voltage (*V_TFL_*), the shallow traps were occupied by injected carriers and the current steeply increased. As the occupied traps formed the space charges, the *I–V* curve followed the SCLC mechanism (slope > 2) [36,39,40]. After the formation of the CFs through the set process, the conduction path represented the LRS and the *I–V* curve followed the linear relationship (slope ~ 1). This LRS was maintained until the reset process, wherein the CFs were ruptured due to the negatively applied bias in Figure 6a. 

### 3.5. Multi-Level Operations and Synaptic Weight Modulation

The memristor devices have received significant attention in recent years because of their use in artificial synapse applications and their analog synaptic weight modulation possibility. The nonvolatile multi-level operations in a single memristor cell can demonstrate the gradual increase and decrease in the conductance in the RS layer. The multi-level resistance properties in a single memristor cell can be analyzed by considering the CFs in the RS layer as a simple cylindrical shape and analytical model during RS operation [41,42]. When the negative bias is applied to the Ti-TE, the oxygen ions that migrate because of the negative electric field recombine with oxygen vacancies and rupture the localized CFs. As the applied voltage for the “reset process” (reset stop voltage, *V_stop_*) increases, further oxygen ion/oxygen vacancy recombination occurs, and the ruptured CFs gap between the TE and BE increases. Therefore, the resistance values of HRS increase with the *V_stop_* during RS operation [43,44]. 

Figure 7 shows the multi-level resistance characteristics for the IGZO memristor devices. Figure 7a shows the multi-level BRS *I–V* characteristic curves obtained by varying the magnitude of *V_stop_* from −1.0 to −1.3 V (in −0.1 V step). In the multi-level BRS *I–V* curves, the read current of HRS and LRS irregularly changed according to the *V_stop_* steps. Figure 7b represents the cumulative probability of multi-level resistance states during 30 cycles of DC BRS operation for each *V_stop_* step. Each resistance value was read at 0.1 V in the BRS *I–V* characteristic curves, and the open and closed symbols represent the distribution of LRS and HRS levels, respectively. In IGZO memristor devices, the probability of resistance was widely distributed, and it was difficult to confirm a clear multi-level according to the *V_stop_* step.

Figure 8 shows the multi-level resistance characteristics for the IGZO:N memristor devices. Figure 8a depicts the multi-level BRS *I–V* characteristic curves by modulating the magnitude of *V_stop_* from −0.7 to −1.2 V (in −0.1 V step). In the multi-level BRS *I–V* curves, the read current of HRS sequentially decreased with the *V_stop_*, which indicates that the CFs gap between the TE and BE also increased. Meanwhile, the read currents of the LRS during the modulation of *V_stop_* were uniform, which indicates that the CFs formations after the “set process” occurred regularly. Figure 8b represents the cumulative probability of multi-level resistance states that were recorded through 30 cycles of DC BRS operation for each *V_stop_* step. Each resistance value was read at 0.1 V in the BRS *I–V* characteristic curves, and the open and closed symbols represent the distribution of LRS and HRS levels, respectively. This indicates the reliable multi-level BRS operations and stable distribution of multi-level resistance according to the *V_stop_* value. Table 4 lists the average resistance values and standard deviations of multi-level states according to the *V_stop_* step. 

Figure 8c shows the retention characteristics of the multi-level resistance states over 10^4^ s at room temperature (25 °C) and high temperature (85 °C), respectively. All resistance states from LRS to HRS exhibited stable nonvolatile retention characteristics, and they did not degrade under room- or high-temperature conditions. Figure 8d represents the relationship between the average resistance values of HRS and the magnitude of the *V_stop_* value. The average resistance values of the HRS can be modulated with the *V_stop_* value, and they demonstrated a linear relationship with a slope of −2.27 dec/*V_stop_* (*R^2^* = 0.995). Therefore, the highly reliable multi-level characteristics of IGZO:N memristor devices suggest the continuous resistance or conductance controllability in the IGZO:N nanocomposites through electrical stimulations [44,45]. 

To investigate the gradual conductance modulation according to the electrical pulse stimulations, which is essential for memristive switching, the synaptic weight increase (potentiation) and decrease (depression) properties were evaluated. Based on the multi-level properties and nonvolatility of the RS nanocomposite, the synaptic weight can be modulated by an implementable algorithm for the repetitive learning process. Figure 9a,c show the conductance increase/decrease characteristics for IGZO and IGZO:N memristor devices for one electrical pulse stimulation cycle, respectively. The one cycle consisted of 50 potentiation pulses (1.2 V/1 ms) and 50 depression pulses (−1.2 V/1 ms), respectively, which were identically repeated. Figure 9a shows the IGZO memristor devices, wherein the conductance increased or decreased irregularly with a dynamic range of ~2 mS during one stimulus cycle. Conversely, in Figure 9c (IGZO:N memristor devices), the conductance increased and decreased symmetrically with a dynamic range of ~6 mS during one stimulus cycle. Figure 9b,d depict the weight modulation stability of IGZO and IGZO:N memristor devices for five consecutive stimulus cycles (a total of 5 × 10^2^ synaptic pulses) evaluations. In Figure 9b, the conductance of IGZO memristor devices was unstably modulated as the pulse number increased. However, in Figure 9d, the conductance of IGZO:N memristor devices was symmetrically and stably potentiated/depressed for the repeated synaptic pulse numbers. As a result, based on the reliable multi-level properties and stable nonvolatility of the IGZO:N nanocomposite, the synaptic weight was successfully modulated in IGZO:N memristor devices, indicating the potential applications in artificial synaptic electronics. 

## 4. Conclusions

We proposed two-terminal memristor devices using solution-derived IGZO:N nanocomposites as an RS layer, and we systematically evaluated the multi-level RS characteristics and artificial synapse operations. The primary outcome of this study was the improvement in memristive switching properties, which was attained by applying microwave-assisted nitridation technology on the solution-derived MeO_x_-based RS layer. The MWA process provided effective volumetric annealing on the solution-derived IGZO layer in a short time through the conversion of electromagnetic energy to thermal energy. As a result, nitrogen species were incorporated into the IGZO nanocomposite framework through microwave-assisted nitridation, which was clearly demonstrated by chemical etching, optical absorption coefficient analysis, and chemical composition analysis using XPS. We prepared two types of memristor devices using an IGZO film or IGZO:N nanocomposite film as RS layers to evaluate the multi-level RS characteristics and electrical synaptic characteristics. It was found that IGZO:N memristors not only had a superior BRS endurance during the 10^3^ DC cycling tests and a stable resistance distribution compared to IGZO memristors, but they also exhibited reliable multi-level BRS characteristics through the modulation of the *V_stop_* value. Furthermore, the multi-resistance values were linearly controlled depending on the magnitude of the *V_stop_* value, and they exhibited stable nonvolatile retention characteristics even under the high-temperature condition. The reliable multi-level properties and nonvolatility of IGZO:N nanocomposites suggest the possibility of gradual synaptic weight modulation through electrical stimulations. Therefore, the conductance in IGZO:N memristors was symmetrically and stably potentiated/depressed for the 5 × 10^2^ repeated synaptic pulse numbers as compared to IGZO memristors. As a result, microwave-assisted nitridation technology is an effective synthesis technology for oxide-based RS layers, and it can improve the memristive switching characteristics for potential synaptic electronics. 

## Figures and Tables

**Figure 1 nanomaterials-11-01081-f001:**
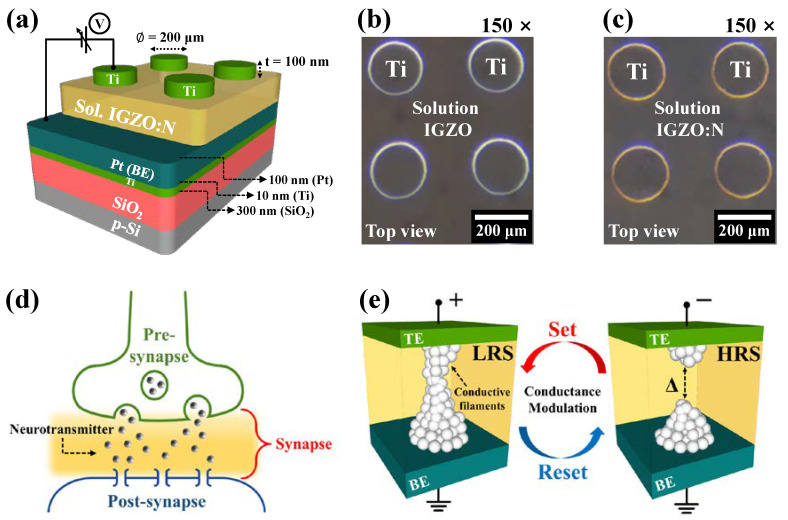
(**a**) Schematic diagram of proposed solution-derived IGZO:N memristors by microwave-assisted nitridation. Optical microscope images (150× magnification) of the memristor devices with MWA treated under an (**b**) air-ambient (IGZO) and (**c**) nitrogen-ambient (IGZO:N). (**d**) Simple structure of the biological synapse. (**e**) Simplified mechanism of conductance modulation through the CFs formation/rupture process.

**Figure 2 nanomaterials-11-01081-f002:**
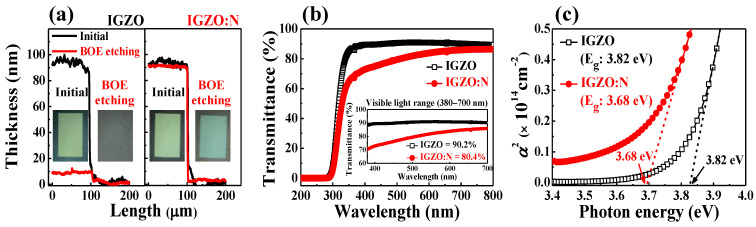
(**a**) Thickness of initial and buffered-oxide-etch (BOE) solution-treated IGZO and IGZO:N films. (**b**) Optical transmittance spectra (inset, optical transmittance at the visible-light wavelength region) and (**c**) optical absorption coefficients of IGZO and IGZO:N nanocomposites.

**Figure 3 nanomaterials-11-01081-f003:**
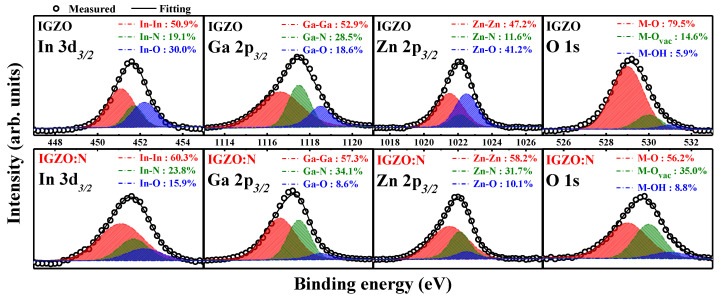
XPS spectra of In *3d*_3/2_, Ga *2p*_3/2_, Zn *2p*_3/2_, and O *1s* peaks in IGZO and IGZO:N nanocomposites.

**Figure 4 nanomaterials-11-01081-f004:**
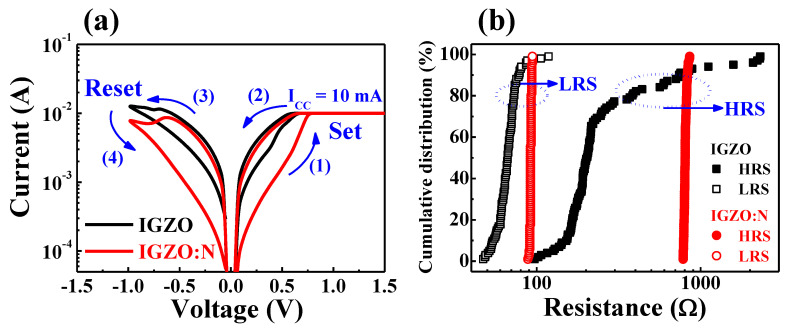
(**a**) Bipolar resistive switching (BRS) *I–V* characteristic curves and (**b**) cumulative resistance distribution under 10^2^ sequential DC cycles of the IGZO and IGZO:N memristor devices.

**Figure 5 nanomaterials-11-01081-f005:**
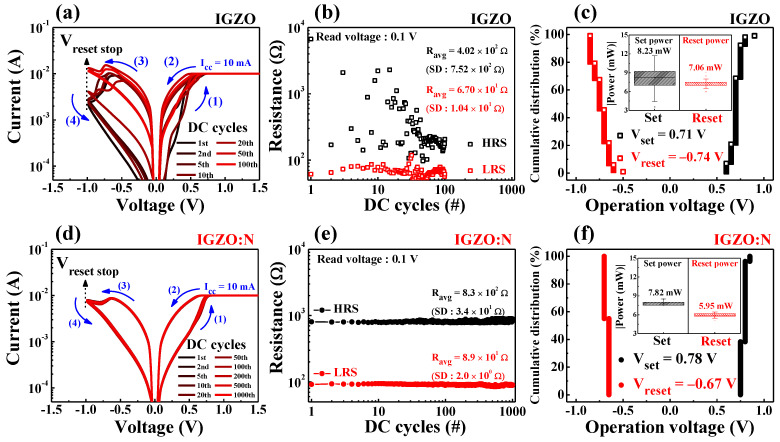
BRS endurance characteristics for (**a**–**c**) IGZO memristors and (**d**–**f**) IGZO:N memristors: (**a**,**d**) BRS *I–V* characteristic curves. (**b**,**e**) The resistance values of HRS and LRS extracted at a read voltage of 0.1 V. (**c**,**f**) Cumulative distribution of set and reset voltages. Insets indicate the calculated operation power for the set and reset processes.

**Figure 6 nanomaterials-11-01081-f006:**
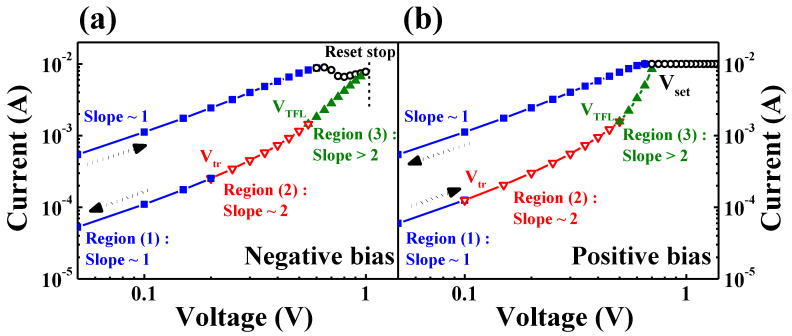
Current conduction mechanism for IGZO:N memristor devices by double-logarithmic-plotted BRS *I–**V* characteristic curves: (**a**) negative bias and (**b**) positive bias regions.

**Figure 7 nanomaterials-11-01081-f007:**
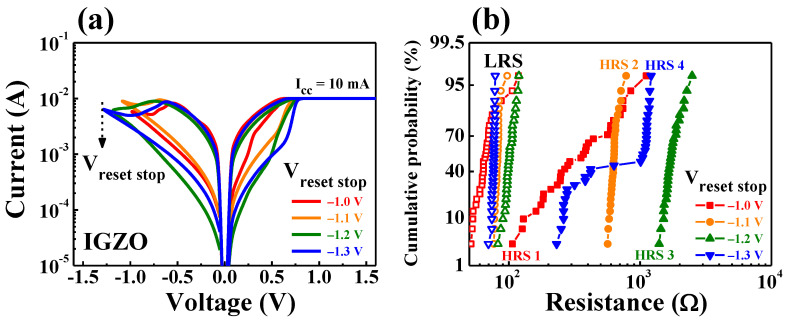
Multi-level resistance characteristics for the IGZO memristor devices by varying the magnitude of *V_stop_* values from −1.0 to −1.3 V (in −0.1 V step). (**a**) BRS *I–V* characteristic curves. (**b**) Cumulative probability during 30 cycles of DC BRS operation.

**Figure 8 nanomaterials-11-01081-f008:**
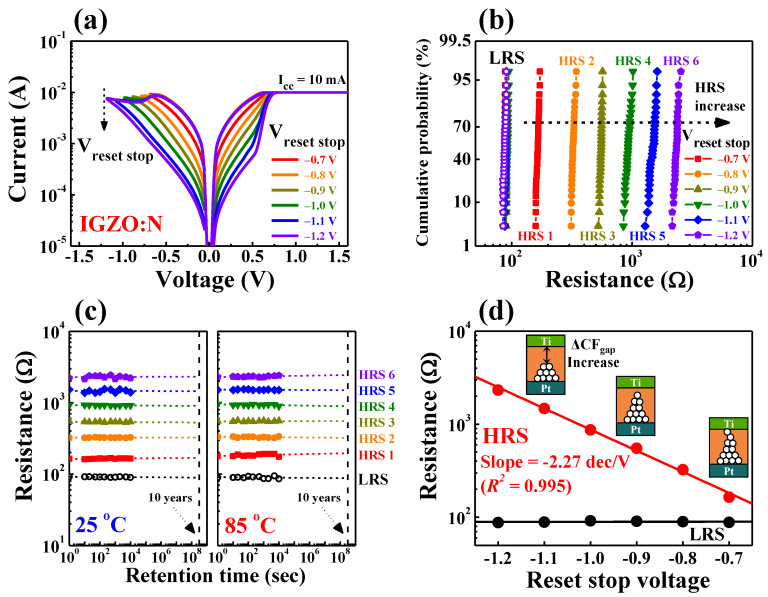
Multi-level resistance characteristics for the IGZO:N memristor devices by modulating the magnitude of *V_stop_* values from −0.7 to −1.2 V (in −0.1 V step). (**a**) BRS *I–V* characteristic curves. (**b**) Cumulative probability recorded through 30 cycles of DC BRS operation. (**c**) Retention characteristics over 10^4^ s at room temperature (25 °C) and high temperature (85 °C). (**d**) Relationship between average resistance values of HRS and magnitude of *V_stop_* value.

**Figure 9 nanomaterials-11-01081-f009:**
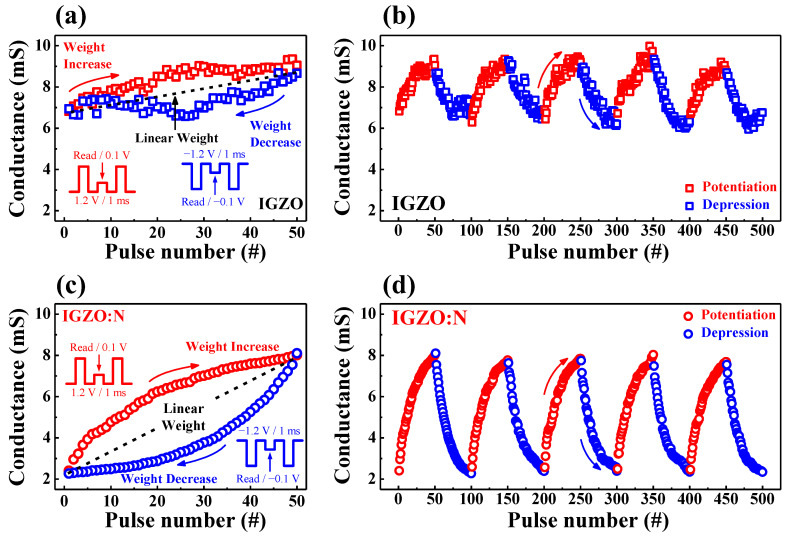
Synaptic weight potentiation/depression for (**a**,**b**) IGZO and (**c**,**d**) IGZO:N memristor devices: (**a**,**c**) Conductance increase and decrease characteristics during one electrical pulse stimulation cycle. (**b**,**d**) Weight modulation stability during five consecutive stimulus cycles (total of 5 × 10^2^ synaptic pulses).

**Table 1 nanomaterials-11-01081-t001:** XPS binding energy values and atomic concentrations for each metal species in IGZO and IGZO:N nanocomposites.

X-ray Photoelectron Spectroscopy										
Orbital		In *3d*_3/2_			Ga *2p*_3/2_			Zn *2p*_3/2_		
Chemical bond		In-In	In-N	In-O	Ga-Ga	Ga-N	Ga-O	Zn-Zn	Zn-N	Zn-O
Binding energy (eV)		451.0	451.5	452.5	1116.6	1117.7	1118.8	1021.7	1021.9	1022.4
Atomic Concentration (%)										
RS layer	IGZO	50.9	19.1	30.0	52.9	28.5	18.6	47.2	11.6	41.2
	IGZO:N	60.3	23.8	15.9	57.3	34.1	8.6	58.2	31.7	10.1

**Table 2 nanomaterials-11-01081-t002:** XPS binding energy values and atomic concentrations for O *1s* peaks in IGZO and IGZO:N nanocomposites.

X-ray Photoelectron Spectroscopy				
Orbital		O *1s*		
Chemical bond		M-O	M-O_vac_	M-OH
Binding energy (eV)		529.0	530.0	531.0
Atomic Concentration (%)				
RS layer	IGZO	79.5	14.6	5.9
	IGZO:N	56.2	35.0	8.8

**Table 3 nanomaterials-11-01081-t003:** Total BRS operation parameters of IGZO and IGZO:N memristor devices.

Device	Parameter	Average (*μ*)	Standard Deviation (*σ*)	*μ ± σ*
IGZO Memristor	Resistance value (HRS)	4.02 × 10^2^ Ω	7.52 × 10^2^ Ω	4.02 × 10^2^ ± 7.52 × 10^2^ Ω
	Resistance value (LRS)	6.70 × 10^1^ Ω	1.04 × 10^1^ Ω	6.70 × 10^1^ ± 1.04 × 10^1^ Ω
	Set voltage (*V_set_*)	0.71 V	0.05 V	0.71 ± 0.05 V
	Reset voltage (*V_reset_*)	−0.74 V	0.08 V	−0.74 ± 0.08 V
	Power for set process (*P_set_*)	8.23 mW	1.96 mW	8.23 ± 1.96 mW
	Power for reset process (*P_reset_*)	7.06 mW	0.51 mW	7.06 ± 0.51 mW
IGZO:N Memristor	Resistance value (HRS)	8.30 × 10^2^ Ω	3.40 × 10^1^ Ω	8.30 × 10^2^ ± 3.40 × 10^1^ Ω
	Resistance value (LRS)	8.95 × 10^1^ Ω	2.01 × 10^0^ Ω	8.95 × 10^1^ ± 2.01 × 10^0^ Ω
	Set voltage (*V_set_*)	0.78 V	0.03 V	0.78 ± 0.03 V
	Reset voltage (*V_reset_*)	−0.67 V	0.02 V	−0.67 ± 0.02 V
	Power for set process (*P_set_*)	7.82 mW	0.27 mW	7.82 ± 0.27 mW
	Power for reset process (*P_reset_*)	5.95 mW	0.24 mW	5.95 ± 0.24 mW

**Table 4 nanomaterials-11-01081-t004:** Average resistance values and standard deviations of multi-level states according to the *V_stop_* step in IGZO:N memristor devices.

Multi-Level State	LRS	HRS 1	HRS 2	HRS 3	HRS 4	HRS 5	HRS 6
Reset stop voltage	-	−0.7 V	−0.8 V	−0.9 V	−1.0 V	−1.1 V	−1.2 V
Average (*μ*)	8.92 × 10^1^ Ω	1.64 × 10^2^ Ω	3.24 × 10^2^ Ω	5.51 × 10^2^ Ω	8.68 × 10^2^ Ω	1.47 × 10^3^ Ω	2.32 × 10^3^ Ω
Standard Deviation (*σ*)	1.77 × 10^0^ Ω	3.94 × 10^0^ Ω	8.77 × 10^0^ Ω	1.13 × 10^1^ Ω	3.57 × 10^1^ Ω	7.73 × 10^1^ Ω	7.78 × 10^1^ Ω

## Data Availability

Not applicable.

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
