# Peer review of "High-Performance Resistive Switching in Solution-Derived IGZO:N Memristors by Microwave-Assisted Nitridation"

_nanomaterials, 2021, doi:10.3390/nano11051081_

Round 1

Reviewer 1 Report

This paper reports on the study of the resistive switching phenomena in solution derived IGZO devices. The work shows that a strongly improvement of the resistive switching performance may be achieved by doping the IGZO films through micro-wave assisted nitrogen annealing. Although this is an interesting study, which may contribute to improve the knowledge on switching devices for memory and neuromorphic applications, I think that there are some important points that may be clarified before considering its publication.

1- The authors compare the switching performance of IGZO and IGZO:N nanocomposites. However, it is not clear from the text if IGZO measurements are performed on as grown samples or on devices MWA-treated in an air-ambient. Is there any influence on the air-ambient annealing as compared with the nitrogen one?

2- The values of Vset and Vreset are almost symmetric for IGZO samples (0.71V / -0.74V) whereas they are rather asymmetric for IGZO:N ones (0.78V/ -0.67V). It would be good if the authors could comment on that point.

3- Nitrogen annealing to improve IGZO-based resistive switching devices has been also studied on films grown by sputtering (Gan et al Vacuum 180 (2020) 109630), https://doi.org/10.1016/j.vacuum.2020.109630). The authors should mention this work and compare their results with it.

Author Response

Response Letter to Reviewers

From Prof. Won-Ju Cho

Department of Electronic Materials Engineering, Kwangwoon University, 20, Gwangun-ro, Nowon-gu, Seoul, 01897, Republic of Korea

Tel: +82-2-940-5163

Fax: +82-2-943-5163

  1. Journal: Nanomaterials
  2. Manuscript ID: nanomaterials-1191674
  3. Title: “High-Performance Resistive Switching in Solution-Derived IGZO:N Memristors by Microwave-Assisted Nitridation”
  4. Authors: Shin-Yi Min, and Won-Ju Cho*

We sincerely thank you for giving us such valuable suggestions for revision. The reviewers′ comments are indeed constructive and helpful for us to improve our manuscript. Thus, we revise our manuscript based on the reviewers′ comments exactly. The followings are our point-to-point response to the reviewers′ concerns and our descriptions on the revision, which are indicated by yellow highlight in the revised manuscript. We hope these revisions would be satisfactory, and this revised manuscript is suitable for publication of Nanomaterials.

Thank you for your kind co-operations.

                                                             Yours sincerely,

                                                             Won-Ju Cho

<Response to Reviewer #1>

[Comment 1]

The authors compare the switching performance of IGZO and IGZO:N nanocomposites. However, it is not clear from the text if IGZO measurements are performed on as grown samples or on devices MWA-treated in an air-ambient. Is there any influence on the air-ambient annealing as compared with the nitrogen one?

[Answer 1]

Thank you very much for your consideration. To systematically evaluate the microwave-assisted nitridation effect on the solution-derived IGZO nanocomposites RS layer, we fabricated and measured both IGZO memristor and IGZO:N memristor devices. The fabrication flow of each device is the same except for the gas ambient during the MWA process. The IGZO memristors and IGZO:N memristor devices were MWA-treated in an air and nitrogen ambient, respectively. As a result, the microwave-assisted nitridation effects were clearly demonstrated by chemical etching, optical absorption coefficient analysis, and X-ray photoelectron spectroscopy. Furthermore, in terms of electrical synaptic operation, the IGZO:N memristors possessed highly improved memristive switching characteristics compared to the MWA-treated in the air ambient devices (IGZO memristors).

In response to the reviewer’s comments, we have also revised the manuscript in the following section for the reader's understanding:

(Line 10 – 13, Page 3 in revised manuscript)

In order to evaluate the microwave-assisted nitridation effect on the memristor operations, the MWA-treated memristor devices in an air-ambient (IGZO memristor devices) were also fabricated and systematically compared with IGZO:N memristor devices.

[Comment 2]

The values of Vset and Vreset are almost symmetric for IGZO samples (0.71V / -0.74V) whereas they are rather asymmetric for IGZO:N ones (0.78V/ -0.67V). It would be good if the authors could comment on that point.

[Answer 2]

Thank you very much for your valuable comments. The average values of Vset and Vreset during BRS endurance properties exhibit more symmetric in the IGZO memristor devices. Nevertheless, in the stability investigation, the standard deviation is more meaningful and important. Table 3 listed the total BRS operation parameters evaluated by the endurance test. As a result, the IGZO:N memristor devices represent highly improved operation stability than IGZO memristor during repeated DC BRS endurance evaluation. Therefore, we further emphasized the improvements in the manuscript.

In response to the reviewer’s comments, we have also revised the manuscript in the following section for the reader's understanding:

(Line 33 – 35, Page 7 in revised manuscript)

Thus, the IGZO:N memristor devices represent highly improved operation stability with small standard deviations during repeated DC BRS endurance evaluation, as compared to IGZO memristors.

[Comment 3]

Nitrogen annealing to improve IGZO-based resistive switching devices has been also studied on films grown by sputtering (Gan et al Vacuum 180 (2020) 109630), https://doi.org/10.1016/j.vacuum.2020.109630). The authors should mention this work and compare their results with it.

[Answer 3]

Thank you for your helpful recommendation. Following the reviewer’s suggestion, we compared this study with previous studies to reinforce the purpose of this study and add the provided reference material.

In response to the reviewer’s comments, we have also revised the manuscript in the following section for the reader's understanding:

(Line 11 – 16, Page 2 in revised manuscript)

Although the IGZO-based RS devices grown by sputtering system have been studied and improved through the conventional thermal annealing (CTA) process in nitrogen-ambient [14]. Compared to the previous study, the solution-based films are promising for versatile electronic applications and the microwave annealing (MWA) technique can offer an effective annealing treatment than the CTA process through its rapid volumetric heating by the conversion of electromagnetic energy into thermal energy [15,16].

Again, thank you very much for your kind consideration and significant advice of our manuscript.

Yours sincerely,

Won­-Ju Cho

Reviewer 2 Report

Authors proposed two-terminal memristor devices using solution-derived IGZO:N nanocomposites as RS layer and evaluated the multi-level RS characteristics and artificial synapse  operations. The primary outcome of their study is the improvement in memristive switching properties attained by applying microwave-assisted nitridation technology on the solution derived MeOx-based RS layer. According to their results, microwave-assisted nitridation technology seems to be an effective synthesis technology for oxide-based RS layers, and it could improve the memristive switching characteristics for potential synaptic electronics.

In my opinion the work is exhaustive, well discussed  and complete with appropriate references. It can be published after minor revisions.

  • References must be numbered in order of appearance in the text. References numeration goes from [24,25] (page 5, line 8) to [29] (page 5, line 20), while [26,27,28] are called in page 6, line 19. Please re-order the References.
  • In Fig. 2 caption, define “BOE”.
  • Page 5, line 2, please define the conduction band “Ec” as you made for valence band, in line 3.
  • Page 4, line 2, “Figures…show” and not “Figures…shows”. Same page, line 30: the sentence is not clear (delete “which”?).
  • In Table 3 and 4 the last column is sufficient, the first two are redundant. Please unify the tables by reporting the data for IGZO and IGZO:N memristor devices in the same table

Author Response

Response Letter to Reviewers

From Prof. Won-Ju Cho

Department of Electronic Materials Engineering, Kwangwoon University, 20, Gwangun-ro, Nowon-gu, Seoul, 01897, Republic of Korea

Tel: +82-2-940-5163

Fax: +82-2-943-5163

  1. Journal: Nanomaterials
  2. Manuscript ID: nanomaterials-1191674
  3. Title: “High-Performance Resistive Switching in Solution-Derived IGZO: N Memristors by Microwave-Assisted Nitridation”
  4. Authors: Shin-Yi Min, and Won-Ju Cho*

We sincerely thank you for giving us such valuable suggestions for revision. The reviewers′ comments are indeed constructive and helpful for us to improve our manuscript. Thus, we revise our manuscript based on the reviewers′ comments exactly. The followings are our point-to-point response to the reviewers′ concerns and our descriptions on the revision, which are indicated by yellow highlight in the revised manuscript. We hope these revisions would be satisfactory, and this revised manuscript is suitable for publication of Nanomaterials.

Thank you for your kind co-operations.

                                                             Yours sincerely,

                                                             Won-Ju Cho

<Response to Reviewer #2>

[Comment 1]

References must be numbered in order of appearance in the text. References numeration goes from [24,25] (page 5, line 8) to [29] (page 5, line 20), while [26,27,28] are called in page 6, line 19. Please re-order the References.

[Answer 1]

Thank you very much for your kind consideration. As reviewer’s comments, we re-ordered the number of references in order.

[Comment 2]

In Fig. 2 caption, define “BOE”.

[Answer 2]

Thank you for your comment. We have defined the “BOE” in Figure 2 caption.

In response to the reviewer’s comments, we have also revised the manuscript in the following section for the reader's understanding:

(Line 20 – 22, Page 4 in revised manuscript)

Figure 2. (a) Thickness of initial and buffered-oxide-etch (BOE) solution-treated IGZO and IGZO:N films. (b) Optical transmittance spectra (inset; optical transmittance at the visible light wavelength region) and (c) optical absorption coefficient of IGZO and IGZO:N nanocomposites.

[Comment 3]

Page 5, line 2, please define the conduction band “Ec” as you made for valence band, in line 3.

[Answer 3]

Thank you for your comment. We have defined the “Ec” to “conduction band (Ec)” like the “valence band (Ev)”.

In response to the reviewer’s comments, we have also revised the manuscript in the following section for the reader's understanding:

(Line 5 – 7, Page 5 in revised manuscript)

In the fundamental bandgap features of IGZO nanocomposites, the conduction band (Ec) is mainly composed of the metal cation related s-orbital, especially In 5 s-orbitals, and the highest valence band (Ev) mainly formed oxygen-related 2p-orbital.

[Comment 4]

Page 4, line 2, “Figures…show” and not “Figures…shows”. Same page, line 30: the sentence is not clear (delete “which”?).

[Answer 4]

Thank you for your accurate point. We have revised the grammatical error and modified an unclear description by deleting “which”.

In response to the reviewer’s comments, we have also revised the manuscript in the following section for the reader's understanding:

(Line 6 – 9, Page 4 in revised manuscript)

Figure 1a depicts a schematic diagram of the proposed solution-derived IGZO:N memristors by microwave-assisted nitridation, and Figures 1b and 1c show the optical microscope images (magnification of 150 ×) of the memristor devices with MWA-treated in air-ambient (IGZO) and nitrogen-ambient (IGZO:N), respectively.

(Line 34 – 36, Page 4 in revised manuscript)

The absorption coefficient is extracted from the ultra-violet wavelength region of optical transmittance spectra and is calculated by the following equation neglecting the reflection coefficient [23]:

[Comment 5]

In Table 3 and 4 the last column is sufficient, the first two are redundant. Please unify the tables by reporting the data for IGZO and IGZO:N memristor devices in the same table.

[Answer 5]

Thank you for your helpful recommendation. For the reader’s better understanding, we have unified the table 3 and 4 in the same table.

In response to the reviewer’s comments, we have also revised the manuscript in the following section for the reader's understanding:

(Line 33 – 35, Page 7 in revised manuscript)

The total BRS operation parameters evaluated by endurance test are listed in Table 3 for IGZO memristors and IGZO:N memristor devices.

(Line 5 – 6, Page 8 in revised manuscript)

Table 3. Total BRS operation parameters of IGZO and IGZO:N memristor devices.

Device

Parameter

Average (μ)

Standard

Deviation (σ)

μ ± σ

IGZO

memristor

Resistance value (HRS)

4.02 × 102 Ω

7.52 × 102 Ω

4.02 × 102 ± 7.52 × 102 Ω

Resistance value (LRS)

6.70 × 101 Ω

1.04 × 101 Ω

6.70 × 101 ± 1.04 × 101 Ω

Set voltage (Vset)

0.71 V

0.05 V

0.71 ± 0.05 V

Reset voltage (Vreset)

-0.74 V

0.08 V

-0.74 ± 0.08 V

Power for set process (Pset)

8.23 mW

1.96 mW

8.23 ± 1.96 mW

Power for reset process (Preset)

7.06 mW

0.51 mW

7.06 ± 0.51 mW

IGZO:N

memristor

Resistance value (HRS)

8.30 × 102 Ω

3.40 × 101 Ω

8.30 × 102 ± 3.40 × 101 Ω

Resistance value (LRS)

8.95 × 101 Ω

2.01 × 100 Ω

8.95 × 101 ± 2.01 × 100 Ω

Set voltage (Vset)

0.78 V

0.03 V

0.78 ± 0.03 V

Reset voltage (Vreset)

-0.67 V

0.02 V

-0.67 ± 0.02 V

Power for set process (Pset)

7.82 mW

0.27 mW

7.82 ± 0.27 mW

Power for reset process (Preset)

5.95 mW

0.24 mW

5.95 ± 0.24 mW

Again, thank you very much for your kind consideration and significant advice of our manuscript.

Yours sincerely,

Won­-Ju Cho

Reviewer 3 Report

Good and interesting article.

I totally agree with the last sentence in the Conclusions section, "As a result, microwave-assisted nitridation technology is an effective synthesis technology for oxide-based RS layers, and can improve memristive switching characteristics for potential synaptic electronics."

Only one addition could improve the article: why does nitridation have such good beneficial effects in stabilizing the overall switching behavior of MIM memristors?

If the authors could give some convincing answer to the latter question, I think the article would be very interesting and could be published as is, in essence.

Author Response

Response Letter to Reviewers

From Prof. Won-Ju Cho

Department of Electronic Materials Engineering, Kwangwoon University, 20, Gwangun-ro, Nowon-gu, Seoul, 01897, Republic of Korea

Tel: +82-2-940-5163

Fax: +82-2-943-5163

  1. Journal: Nanomaterials
  2. Manuscript ID: nanomaterials-1191674
  3. Title: “High-Performance Resistive Switching in Solution-Derived IGZO: N Memristors by Microwave-Assisted Nitridation”
  4. Authors: Shin-Yi Min, and Won-Ju Cho*

We sincerely thank you for giving us such valuable suggestions for revision. The reviewers′ comments are indeed constructive and helpful for us to improve our manuscript. Thus, we revise our manuscript based on the reviewers′ comments exactly. The followings are our point-to-point response to the reviewers′ concerns and our descriptions on the revision, which are indicated by yellow highlight in the revised manuscript. We hope these revisions would be satisfactory, and this revised manuscript is suitable for publication of Nanomaterials.

Thank you for your kind co-operations.

                                                             Yours sincerely,

                                                             Won-Ju Cho

<Response to Reviewer #3>

[Comment 1]

Only one addition could improve the article: why does nitridation have such good beneficial effects in stabilizing the overall switching behavior of MIM memristors?

[Answer 1]

Thank you for your helpful consideration. The overall memristive switching characteristics in IGZO:N memristor devices are improved and stabilized by microwave-assisted nitridation effects. In the chemical compositions of IGZO and IGZO:N nanocomposites by X-ray photoelectron spectroscopy, the atomic concentration of M-Ovac bonds significantly increases in IGZO:N nanocomposites. This evidence indicates the incorporation of nitrogen species into the IGZO nanocomposites by the microwave-assisted nitridation process, which could result in sufficient oxygen-vacancies in the RS layer. Therefore, the oxygen-vacancies rich CFs in IGZO:N RS layer can reinforce the switching behavior stability and potential synaptic weight modulation reliability.

In response to the reviewer’s comments, we have also revised the manuscript in the following section for the reader's understanding:

(Line 35, Page7 – Line 4, Page 8 in revised manuscript)

Thus, the IGZO:N memristor devices represent highly improved operation stability with small standard deviations during repeated DC BRS endurance evaluation, as compared to IGZO memristors. These beneficial switching behavior stabilization effects are due to the oxygen-vacancies rich CFs in IGZO:N nanocomposites which are attributed by microwave-assisted nitridation [12,34,35].

Again, thank you very much for your kind consideration and significant advice of our manuscript.

Yours sincerely,

Won­-Ju Cho

Round 2

Reviewer 1 Report

The authors have performed all the suggested corrections and thus I think that the paper is ready for publication.